# TASK VECTORS ARE CROSS-MODAL

**(a) Same Task, Different Specifications**

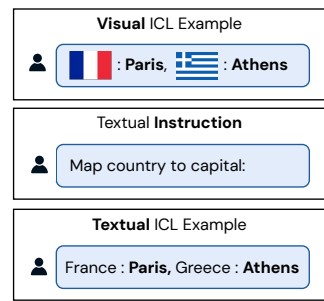

**(b) The Embedding Space of Task Representations**

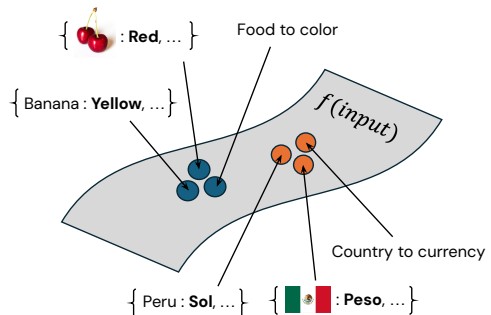

Figure 1: Modern autoregressive vision-and-language models (VLMs) are quite flexible; they can execute the same task expressed in various ways (a). We find that VLMs map these diverse inputs to similar task representations, across modalities and specifications (b).

## ABSTRACT

We investigate the internal representations of autoregressive vision-and-language models (VLMs) and how they encode task representations. We consider tasks specified through examples or instructions, using either text or image inputs. Surprisingly, we find that conceptually similar tasks are mapped to similar task vector representations, regardless of how they are specified. Our findings suggest that to output answers, tokens in VLMs undergo three distinct phases: input, task, and answer, a process which is consistent across different modalities and specifications. The task vectors we identify in VLMs are general enough to be derived in one modality (e.g., text) and transferred to another (e.g., image). Additionally, we find that ensembling exemplar and instruction based task vectors produce better task representations. Taken together, these insights shed light on the underlying mechanisms of VLMs, particularly their ability to represent tasks in a shared manner across different modalities and task specifications.

## 1 INTRODUCTION

Many modern vision-and-language models (VLMs) are designed as autoregressive models that tackle various computer vision tasks through text. For example, tasks like image recognition, OCR, and object detection can be formulated as visual question answering (Antol et al., 2015) and solved with textual outputs (Alayrac et al., 2022; Lu et al., 2022; Liu et al., 2023a).

Despite their success, the underlying structures and inductive biases that drive such VLMs remain a mystery. This urges us to ask what representations enable VLMs to process multi-modal inputs to answer questions. We investigate a specific type of representation known as task vectors, which have been studied in language-only (Hendel et al., 2023; Todd et al., 2024) and vision-only models (Hojel et al., 2024). These studies observe that models conditioned on in-context learning (ICL) examples contain token representations that encode task information.

In this work, we discover that VLMs encode tasks within a shared embedding space, where similar tasks are clustered together regardless of how they are specified. We examine tasks that can be defined through either text or image examples, as well as instructions. For instance, the task of

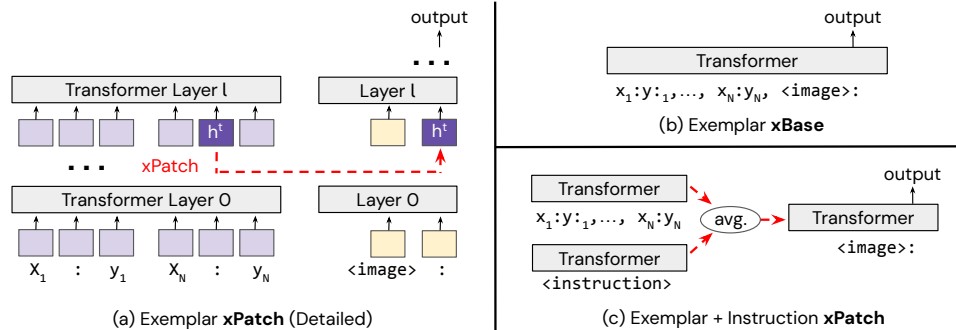

Figure 2: **Cross-modal transfer**. Task vectors can be patched cross-modally (a), outperforming the few-shot prompting baseline (b). We find that task vectors can also be instantiated with instructions, which can be averaged with exemplar-based vectors to produce more stable task representations (c).

mapping a country to its capital (see Figure 1a) can be expressed using text examples (e.g., "France: Paris"), explicit instructions ("Map country to capital"), or image-text pairs (e.g., an image of the French flag labeled "Paris"), all of which result in similar task representations (see Figure 1b). A corresponding t-SNE visualization is provided in Sec. A.7 of the Appendix.

More specifically, we investigate task vectors in VLMs and demonstrate that they are *cross-modal*, allowing task representations to transfer between modalities (see Figure 2). Our analysis further reveals that as VLMs generate answers, token representations evolve across model layers in a consistent pattern: starting with the literal input, transitioning to the task representation, and finally, converging to an answer. This suggests that not only are task representations similar across modalities but the entire process of answer generation may be shared, despite differences in task specification.

Motivated by this similarity between the token representations regardless of the input modality, we quantitatively evaluate the cross-modal transfer performance of task vectors for early-fusion and late-fusion VLMs on a range of tasks. For text-to-image transfer, cross-modal patching can improve over text ICL in the same context window by as much as 33%. Ensembling text instructions with examples can improve the sample efficiency of the task vector, with an 18% performance improvement over examples alone in the low-data regime. Surprisingly, we also find that task vectors are transferable between the base LLM and the fine-tuned VLM, meaning that the VLM is able to re-purpose functions learned in a language-only setting on image queries.

Our contributions are threefold. First, we illustrate a taxonomy of task vectors, where they can be specified not only via examples as studied in prior work but also instructions. Second, we show that VLM representations evolve in a common pattern regardless of the input modality or specification format. Finally, we explore cross-modal transfer, which is a useful measure for the interchangeability of different task representations and offers greater expressiveness when defining tasks.

## 2 CROSS-MODAL TASK VECTORS

In Sec. 2.1, we review preliminaries, followed by a discussion in Sec. 2.2 on how task vectors can be specified and transferred in VLMs. Finally, in Sec. 2.3 we explore how the output representations evolve, explaining why cross-modal transfer is feasible.

### 2.1 TASK VECTOR PATCHING PRELIMINARIES

In-context learning can be formulated as follows. For a given task $t \in \mathcal{T}$, a few-shot prompt can be constructed from $N$ input-output examples $p^t = [(x_1, y_1), \ldots, (x_N, y_N)]$. The model $f$ has to learn the mapping from input to output from $p^t$ and apply it onto $x_q$. Previous work has shown that large transformer models implicitly compress this function into a latent activation, also called the *task vector*, for both LLMs (Hendel et al., 2023; Todd et al., 2024) and computer vision models (Hojel et al., 2024). Specifically, the forward pass $f(p^t)$ produces intermediate latent activations that capture the task information, in some transformer layer $l \in L$ at the delimiter token between the last input and output $(x_N, y_N)$. Thus, the original function can be decomposed into the task vector (a

Table 1: **Cross-modal tasks.** We design six tasks inspired by the text examples in prior work (Hendel et al., 2023; Todd et al., 2024), where we add alternative specifications such as instructions and image examples. We provide more details in Sec. A.1 of the Appendix.

| Task | Instruction | Text ICL Example | Image ICL Example |
|---|---|---|---|
| Country-Capital | *The capital city of the country:* | {Greece : **Athens**} | {  : **Athens**} |
| Country-Currency | *The last word of the official currency of the country:* | {Italy : **Euro**} | {  : **Euro**} |
| Animal-Latin | *The scientific name of the animal's species in latin:* | {Gray Wolf : **Canis lupus**} | {  : **Canis lupus**} |
| Animal-Young | *The term for the baby of the animal:* | {Common Dolphin : **calf**} | {  : **calf**} |
| Food-Color | *The color of the food:* | {Persimmon : **orange**} | {  : **orange**} |
| Food-Flavor | *The flavor descriptor of the food:* | {Strawberry : **sweet**} | {  : **sweet**} |

forward pass producing $h^t$) and the query (a forward pass with only $x_q$ and no task information):

$$h^t = f_l(p^t) \qquad y_q = f(x_q \mid h^t) \tag{1}$$

where $h^t$ denotes the intermediate output of the $l$-th transformer layer at the last delimiter token, and $f(x_q \mid h^t)$ denotes task vector patching onto the contextless query at the layer and token corresponding to $h^t$. For autoregressive models, i.e., the LLMs studied in prior work and the VLMs we study, $f(p_t)$ represents a distribution for the next token prediction. We hypothesize that VLMs also encode task vectors in their activation space during the forward pass, which we discuss next.

## 2.2 CROSS-MODAL PATCHING

Our main finding is that task vectors are cross-modal and remain consistent despite different specifications, and therefore can be transferred. Given a task $t \in \mathcal{T}$, we explore three different specification formats: textual exemplars, image exemplars, and textual instructions. We construct six evaluation tasks, where we display these analogous specifications in Table 1. In this work, we categorize settings by cross-modality (denoted by the modifier x) and application method (either prompting, Base, or patching, Patch). Thus, our proposed cross-modal patching method is referred to as xPatch.

**Method.** In Figure 2a, we illustrate one case of cross-modal patching. Here we patch from textual exemplars onto an image query. We run two forward passes: one to extract the task vector from the exemplars and one with a contextless query. We extract the task vector $h^t$ from the $l$-th transformer layer output at the delimiter token between the last input-output pair $(x_N, y_N)$, and we inject it directly at the corresponding layer and token position of the query. To obtain a good estimate of $h^t$, we sample and average the activations from multiple task prompts, and we determine the best layer $l$ for each model via average task accuracy on the validation set. We also compare against the few-shot prompting baseline, where the task specification and query are jointly fed to the transformer, see Figure 2b. We explore three main cases of cross modal patching, corresponding to the different specification formats, which we formalize below.

**Text ICL Transfer.** A task vector from text examples $p_{txt}^t$ can be patched onto image query $x_{img}$.

$$h_{txt}^t = f_l(p_{txt}^t) \qquad y_{img} = f(x_{img} | h_{txt}^t) \tag{2}$$

We refer to this setting as *Text ICL xPatch*. We also look at a special case transferring task vectors from a base LLM to its fine-tuned VLM, which we call *LLM-VLM xPatch*.

**Instruction Transfer.** A task vector from instruction $p_{inst}^t$ can be patched onto image query $x_{img}$.

$$h_{inst}^t = f_l(p_{inst}^t) \qquad y_{img} = f(x_{img} | h_{inst}^t) \tag{3}$$

While prior work only studies exemplars, we also consider instructions, which are more direct and require no input-output samples. We explore the utility of such instructions for making exemplar-

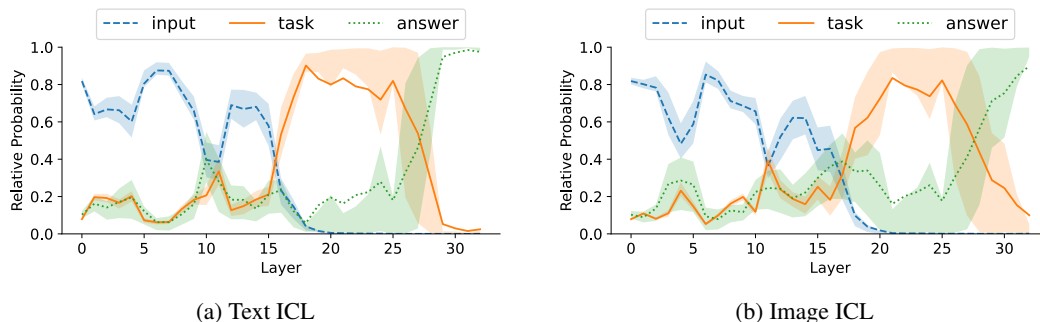

(a) Text ICL                                    (b) Image ICL

Figure 3: **The output evolves in three distinct phases that are shared for text and image ICL**. Each line corresponds to the probability that the last token representation decodes to a pre-defined input, task, or answer vector. We display visualizations of specific layers in Figure 4 and further visualize the task representation phase in Table 2.

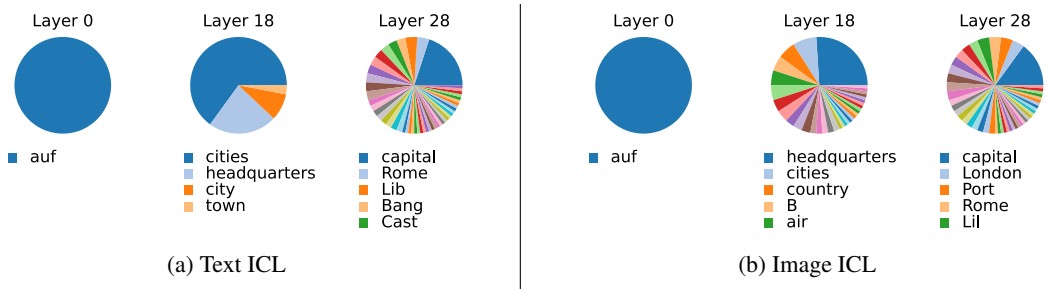

(a) Text ICL                                    (b) Image ICL

Figure 4: **The output transforms from input to task to answer across model layers**. Each pie chart slice represents a top-1 decoding across 100 sets of ICL examples for the Country-Capital task, with the most common decodings below.

Table 2: **The task vector, whether textual or visual, often decodes to task summaries.** The table depicts the top-5 decodings for each task, where $\Diamond$ denotes non-word tokens.

| Task | Text ICL | Image ICL |
|---|---|---|
| Country-Capital | *headquarters, cities, city, cidade, centro* | *headquarters, administr, cities, city, $\Diamond$* |
| Country-Currency | *currency, currency, dollar, dollars, Currency* | *currency, $\Diamond$, currency, undefined, dollars* |
| Animal-Latin | *species, genus, habitat, mamm, american* | *species, genus, mamm, spec, creature* |
| Animal-Young | *pup, babies, baby, called, young* | *young, species, scriptstyle, animal, teenager* |
| Food-Color | *yellow, pink, green, purple, orange* | *green, yes, yellow, verd, yes* |
| Food-Flavor | *flavor, taste, mild, flav, tastes* | *yes, none, anger, cerca, vegetables* |

based task vectors more robust, denoted as *Exemplar + Instruction xPatch* (see Figure 2c). We also look at a scenario of conflicting instructions, denoted as *Instruction xBase vs. Instruction xPatch*.

**Image ICL Transfer.** A task from image examples $p_{img}^t$ can be patched onto text query $x_{txt}$.

$$h_{img}^t = f_l(p_{img}^t) \qquad\qquad y_{txt} = f(x_{txt}|h_{img}^t) \qquad\qquad (4)$$

We refer to this setting as *Image ICL xPatch*. We find that image ICL can be useful for tasks that map a dense textual description to its underlying visual concept.

## 2.3 TOKEN REPRESENTATION EVOLUTION

We investigate how token representations evolve to generate answers. Our main finding is that tokens evolve similarly regardless of whether the ICL queries are expressed via text or image. We start by analyzing how tokens evolve during ICL then focus on the "task" phase, where the task representation emerges. We also include a similar analysis for instructions in Sec. A.7 of the Appendix.

Table 3: **Cross-modal transfer results**. We display the accuracy across six tasks on an unseen test set. For image queries, patching cross-modal task vectors (Text ICL xPatch) outperforms text ICL in the same context window (Text ICL xBase) and the strong unimodal image ICL baseline (Image ICL Base, Patch). The best method per task is underlined and overall is **bolded**.

| Model | Country-Capital | Country-Currency | Animal-Latin | Animal-Young | Food-Color | Food-Flavor | Avg. |
|---|---|---|---|---|---|---|---|
| Random | 0.00 | 0.12 | 0.00 | 0.18 | 0.24 | 0.31 | 0.14 |
| **LLaVA-v1.5** | | | | | | | |
| No Context | 0.00 | 0.00 | 0.00 | 0.00 | 0.00 | 0.00 | 0.00 |
| Image ICL Base | - | - | - | - | - | - | - |
| Image ICL Patch | - | - | - | - | - | - | - |
| Text ICL xBase | 0.02 | 0.18 | 0.03 | 0.23 | 0.28 | 0.37 | 0.18 |
| Text ICL xPatch | 0.31 | 0.30 | 0.26 | 0.18 | 0.53 | 0.31 | **0.32** |
| **Mantis-Fuyu** | | | | | | | |
| No Context | 0.00 | 0.00 | 0.00 | 0.00 | 0.00 | 0.00 | 0.00 |
| Image ICL Base | 0.11 | 0.13 | 0.24 | 0.05 | 0.34 | 0.23 | 0.18 |
| Image ICL Patch | 0.17 | 0.03 | 0.16 | 0.05 | 0.50 | 0.31 | 0.20 |
| Text ICL xBase | 0.09 | 0.06 | 0.08 | 0.02 | 0.23 | 0.04 | 0.09 |
| Text ICL xPatch | 0.32 | 0.23 | 0.36 | 0.09 | 0.51 | 0.36 | **0.31** |
| **Idefics2** | | | | | | | |
| No Context | 0.03 | 0.00 | 0.03 | 0.00 | 0.01 | 0.01 | 0.01 |
| Image ICL Base | 0.71 | 0.57 | 0.43 | 0.12 | 0.41 | 0.35 | 0.43 |
| Image ICL Patch | 0.58 | 0.32 | 0.40 | 0.03 | 0.39 | 0.17 | 0.31 |
| Text ICL xBase | 0.11 | 0.03 | 0.41 | 0.13 | 0.21 | 0.18 | 0.18 |
| Text ICL xPatch | 0.61 | 0.40 | 0.48 | 0.62 | 0.53 | 0.39 | **0.51** |

**Identifying Three Phases.** We first look at all the phases the token representation undergoes across model layers. We analyze Idefics2 (Laurençon et al., 2024), which supports both text and image ICL. Using logit lens (nostalgebraist, 2020), we leverage the model's existing vocabulary space to decode the last token representation. In Figure 3 we visualize the probability the token decodes to these different embedding types (input, task, and answer), where we define the tokens in each category manually per task. In Figure 4 we dive into individual phases, showing the set of top-1 decodings for different model layers. The early layer decodes to the token *auf*, which in Idefics2 globally corresponds to the colon, or the input used for the last token. The middle layer decodes to a small set of task summaries similar to those displayed in Table 2. The late layer decodes to tokens that resemble the output space. We limit the visualization in both figures to the Country-Capital task and provide visualizations for all tasks in Sec. A.7 of the Appendix.

**Decoding the Task Phase.** Drilling down to the task phase, we take the token representation at a middle layer and average it across multiple runs, then depict the top-5 decodings in Table 2. We find that task vectors defined in either modality often decode into meta-tokens that summarize the task. The text-only case is consistent with prior work (Hendel et al., 2023; Todd et al., 2024) that investigates such decodings in language models. For example *headquarters*, *currency*, and *species* are the top-1 decodings for both text and image ICL in the first three tasks in the table. In the case of image ICL, this alignment with language is not immediately obvious. Prior work has shown the input image and text embeddings are quite different, i.e., these embeddings exhibit low cosine similarity (Lin et al., 2024) and form distinct PCA clusters (Liang et al., 2024). Even more, the decodings for image ICL are often noisier than text ICL, which suggests that cross-modal patching could help convey a cleaner expression of the task.

## 3 EXPERIMENTS AND RESULTS

Next, we evaluate the cross-modal transfer performance of task vectors derived from different specifications. In Sec. 3.1 we evaluate the transfer performance from text ICL to image queries, including the inter-model case of LLM to VLM transfer. In Sec. 3.2 we demonstrate that instruction-based vectors can be ensembled with exemplar-based vectors and override pre-existing instructions. In Sec. 3.3 we show qualitative examples where image ICL benefits text queries.

**Models.** We evaluate on three models which represent a broad spectrum of architectures prevalent within modern VLMs. LLaVA-v1.5 (Liu et al., 2024) is a late-fusion model that fine-tunes a projection from visual features into the representation space of a language model. Mantis-Fuyu (Bavishi et al., 2023; Jiang et al., 2024) is an instruction-tuned variant of an early-fusion transformer trained to jointly handle image and text inputs from scratch, where the "visual encoder" is a linear projection on top of the raw image patches. Idefics2 (Laurençon et al., 2024) is a late-fusion model optimized

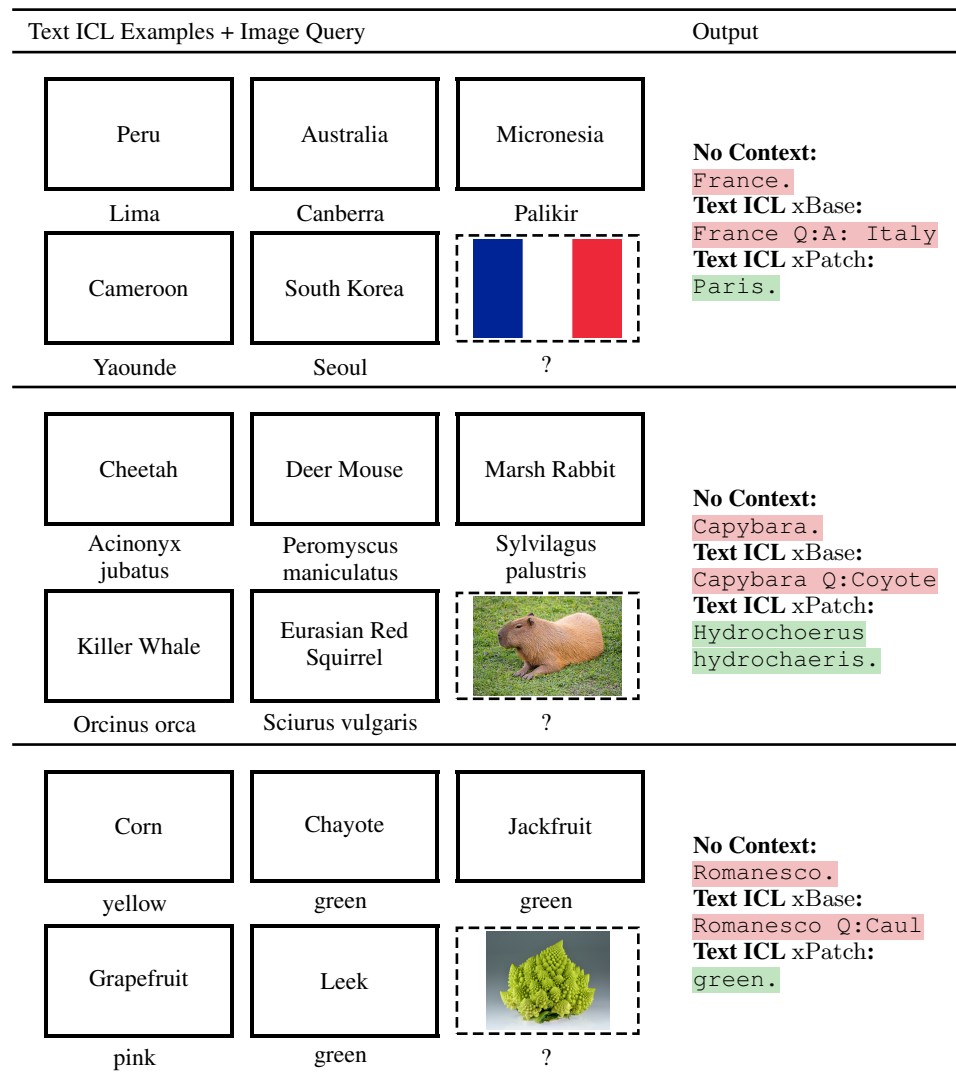

Figure 5: **Transfer from text ICL to image queries**. We show qualitative examples, where few-shot prompting with text ICL (xBase) regurgitates the input while cross-modal patching (xPatch) successfully performs the task.

for multimodal in-context learning, as it aggressively compresses visual features and trains on interleaved image-text documents. We provide more model details in Table 5 of the Appendix.

**Baselines.** To evaluate whether cross-modal task vectors are useful (xPatch), we compare against several baselines. We ablate cross-modality by comparing with the unimodal baselines (Base and Patch), and we ablate the application method by comparing against few shot-prompting with cross-modal examples (xBase). We also compute the performance of two lower bounds – the majority answer from ICL examples (Random) and the query without any task information (No Context).

**Experimental Setup.** For all models, we use the generic template from Todd et al. (2024):

$$\texttt{Q:}\{x_1\}\texttt{\textbackslash nA:}\{y_1\}\texttt{\textbackslash n\textbackslash n}\cdots\texttt{Q:}\{x_n\}\texttt{\textbackslash nA:}\{y_n\}$$

where we evaluate with $N = 5$ ICL examples. For every task, we use 30 samples for validation and 100 samples for testing. We report metrics on the unseen test set, averaged over three seeds. When computing accuracy metrics, we follow prior work (Hendel et al., 2023; Todd et al., 2024) and compare whether the first generated token is an exact match with the pre-defined label. We resize all images to a standard width of 224 pixels. All additional examples and results correspond to Idefics2, the best performing model, unless otherwise specified.

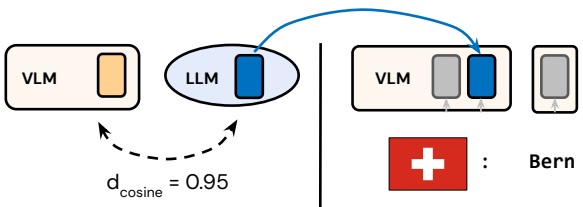

Figure 6: **Inter-model transfer.** For the same text ICL inputs, the base LLM and fine-tuned VLM contain highly similar task vectors (left). LLM task vectors can be patched onto image queries (right).

Table 4: **LLM to VLM transfer results**. We display the cosine similarity between the text ICL task vectors of both models and the test accuracy patching from text ICL in the LLM to image queries in the VLM.

| Model | Cosine Sim. | Avg. |
|---|---|---|
| Random | 0.58 | 0.14 |
| **LLaVA-v1.5** | | |
| VLM-VLM xPatch | - | 0.32 |
| LLM-VLM xPatch | 0.95 | **0.37** |
| **Idefics2** | | |
| VLM-VLM xPatch | - | 0.51 |
| LLM-VLM xPatch | 0.89 | **0.52** |

## 3.1 TEXT ICL TRANSFER

**Quantitative Evaluation.** Recall Sec. 2, where we observe that whether the same task is represented via text or image samples, the model compresses these demonstrations into interpretable task vectors. With this in mind, can we provide demonstrations using only text and apply them to an image query? We evaluate this transfer setting in Table 3 and show qualitative results in Figure 5.

We find that cross-modal patching performs the best across all VLMs (Text ICL xPatch). Patching performs 14-33% better than providing the examples in the same context window (Text ICL xBase). In fact, Text ICL xBase struggles to even execute the task on the image query, which performs at most 4% better than Random. One possible explanation is that mixed-modal examples are relatively out-of-domain whereas decomposed task vectors are more in-domain for the model.

The cross-modal text examples are more helpful than the unimodal image examples, with Text ICL xPatch outperforming the strongest image ICL baseline (Image ICL Base, Patch) by 8-13%. We hypothesize that image ICL requires an additional visual recognition step to understand the task compared with text ICL, which may lead to noisier task representations (see Table 2).

**LLM to VLM Transfer.** Given that many VLMs are initialized from a pre-trained LLM, we explore the extent to which the task representations are preserved after fine-tuning. We illustrate the transfer setting for the base LLM task vectors in Figure 6 and report quantitative results in Table 4. We limit this evaluation to the late-fusion models with a corresponding LLM, where LLaVA-v1.5 corresponds to Vicuna (Chiang et al., 2023) and Idefics2 corresponds to Mistral (Jiang et al., 2023).

We find that given the same text ICL examples, the base LLM and VLM produce highly similar task vectors. The task vectors have a cosine similarity of 0.89 or more, which is much higher than the random baseline which averages the cosine similarity between all mismatched pairings of task vectors in Idefics2. Motivated by this observation, rather than transferring text ICL task vectors to image queries in the same model (VLM-VLM xPatch), we evaluate inter-modal transfer (LLM-VLM xPatch). Surprisingly, the LLM-VLM setting performs 1-5% better than the VLM-VLM setting. This result suggests VLMs can reuse functions learned only in language by LLMs, and that some elements of the base LLM's task representation space may be retained after fine-tuning.

## 3.2 INSTRUCTION TRANSFER

In Sec. 2.2 we proposed instruction-based task vectors, which are defined directly via textual instruction. We illustrate the effect of patching instruction-based vectors onto image queries in Figure 7.

**Complementarity with Examples.** We explore whether instruction- and exemplar-based vectors can be combined to produce better task representations in Figure 8. To begin, we evaluate how the test performance scales with the number of ICL examples by computing per-task exemplar-based vectors on subsets of the validation set (Exemplar xPatch). Next, we average the per-task instruction-based vector with each exemplar-based vector (Instruction + Exemplar xPatch). We also plot the performance of the lone instruction-based vector for reference (Instruction xPatch). Because it is difficult to illustrate the desired casing style using only instructions, in this figure only we compute accuracy metrics in a case-insensitive fashion.

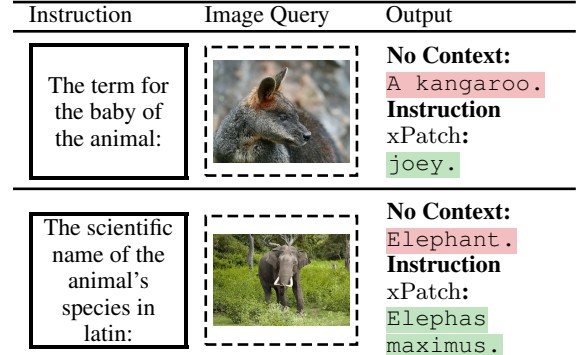

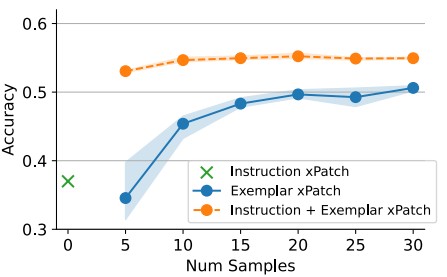

Figure 7: **Instruction-Based Vectors.** Task vectors can also be defined via brief instructions and patched onto image queries (Instruction xPatch).

Figure 8: **Vector Ensembling**. Averaging textual instruction- and exemplar-based vectors improves sample efficiency. We display the number of input-output samples used versus average test accuracy for cross-modal patching onto image queries.

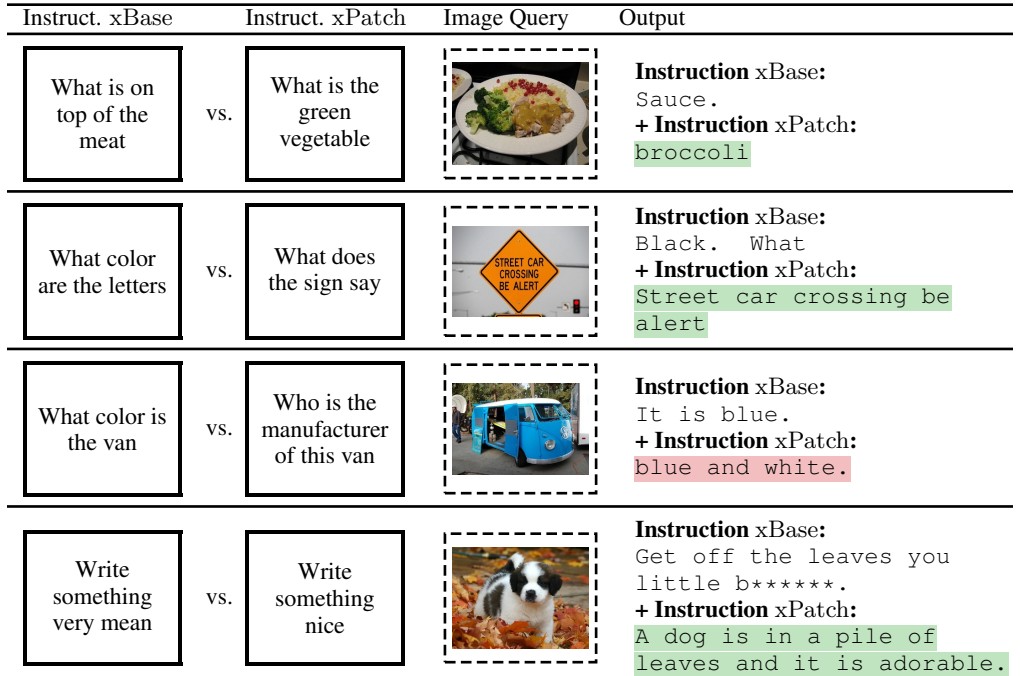

Figure 9: **Task conflict.** We show qualitative examples where the task specified in the same context window (xBase) conflicts with the task to patch (xPatch). Any offensive text has been redacted.

Viewing Figure 8, although the instruction-based vector has not seen any input-output pairs, it shows competitive patching performance, matching that of an exemplar-based vector composed of five samples. The ensemble performs even better, improving over the five-sample exemplar-based vector by 18%. Overall, combining the instruction-based vector improves the sample efficiency and reduces the variance of the exemplar-based vector. We hypothesize that the ensemble performs well because the instruction provides a generic task definition less biased by the selection of input-output examples while the ICL examples provide a sense of the expected output format.

**Task Conflict.** In Figure 9 we consider a special case of cross-modal patching where the task to patch conflicts with an existing task given in the prompt. This case mirrors a practical challenge where the user may request a task that goes against the global system instruction. We give the model conflicting question answering tasks (Goyal et al., 2017), as well as a scenario where the user prompts for toxicity, which conflicts with the patched system instruction. We first display the result where only one task is prompted within the context window (Instruction xBase). We then display the result when the conflicting task is patched on top (+ Instruction xPatch).

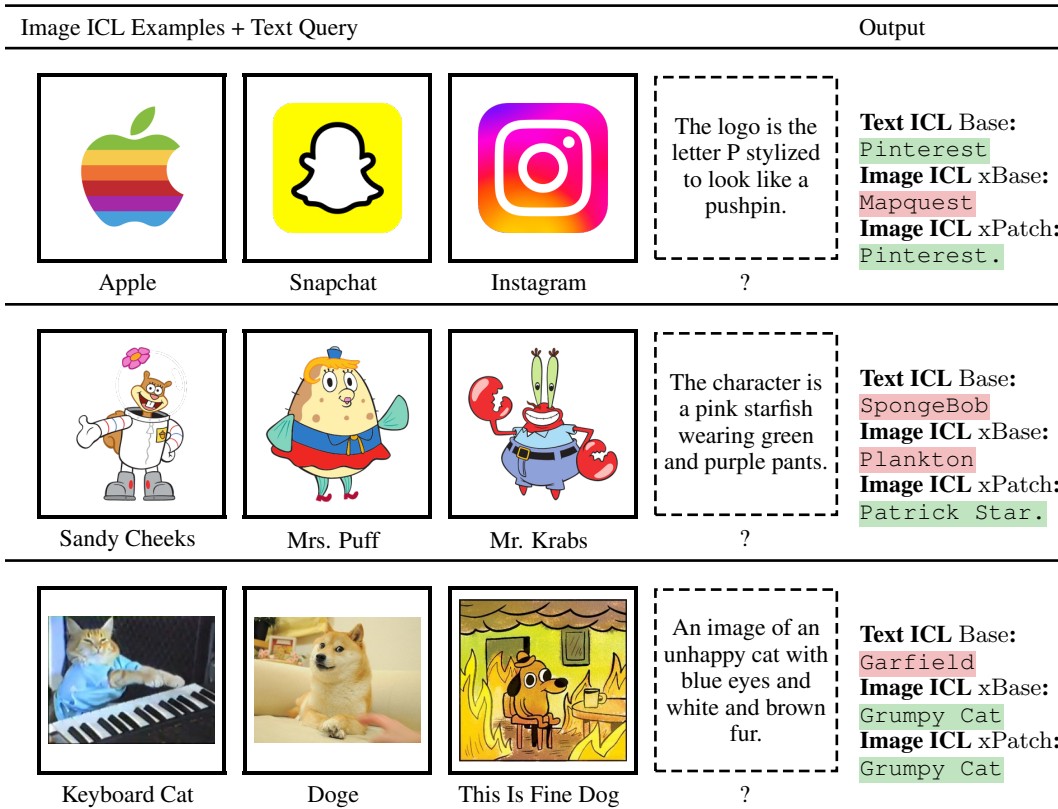

Figure 10: **Transfer from image ICL to text queries**. We show qualitative examples where few-shot prompting with text ICL (Base) and image ICL (xBase) often produces incorrect predictions in the same output domain while cross-modal patching (xPatch) leads to the correct answer.

We observe that global vector patching is often able to override local prompting but also fails when the task to patch is more challenging than the one provided in the same context window. For example, tasks like object recognition, color identification, or OCR that are highly emphasized in VLM training can be considered less challenging than a long-tail task like car logo recognition.

### 3.3 IMAGE ICL TRANSFER

Now we assess the usefulness of task vectors derived from image ICL for text queries, as originally formulated in Sec. 2.2. In Figure 10 we depict a set of tasks that involve recognizing visual concepts in dense textual descriptions, including mapping the description to a technology company, cartoon character, or popular meme. We provide the text ICL descriptions in Sec. A.6 of the Appendix.

Similar to Sec. 3.1, the model struggles when cross-modal examples are applied via few-shot prompting (Image ICL xBase) but performs well when the same examples are patched as a task vector (Image ICL xPatch). Both baselines (Text ICL Base, Image ICL xBase) sometimes generate incorrect answers within the same output domain, suggesting that, rather than focusing on the input-output relationship, the model may be ignoring the input image or description. However, on the evaluation tasks in Table 3, it is difficult for image ICL to surpass the strong unimodal baselines. In Table 10 of the Appendix we include an ablation containing all possible combinations of specification-query modality for task vector patching, where text ICL consistently outperforms image ICL regardless of the query modality. We hypothesize that this phenomenon can be attributed to the nature of the tasks themselves. In the evaluation tasks, image ICL also has to complete an implicit recognition task mapping the image to the underlying textual concept. For example, if the model cannot match the flag to the correct country name, it will not be able to predict the correct currency. However, if recognition is instead required in text space, as is the case in Figure 10, image ICL may better encode the task. We think that the curation of a comprehensive evaluation set containing dense text descriptions and corresponding visual concepts is an exciting future direction.

## 4    RELATED WORK

**Mechanistic Interpretability.** The goal of mechanistic interpretability in deep learning is to make deep models more transparent and interpretable by understanding how and why model decisions are made (Gilpin et al., 2018; Gurnee & Tegmark; Liu et al., 2022; Geva et al., 2020; Nanda et al., 2023). To uncover the relationships within the model, *causal interventions* (Pearl, 2022) are often used. For example, Activation Patching (Zhang & Nanda, 2023) is a technique used to modify neural network activations to observe changes in outputs, often with causal insights to correct biased or erroneous behavior (Meng et al., 2022; Bau et al.). Here, we use Activation Patching to demonstrate that task representations transfer across modalities, regardless of being specified by examples or instructions.

**In Context Learning.** With the recent advent of LLMs (Brown et al., 2020), researchers have sought to explain in-context learning (Liu et al., 2023b), the phenomenon in which LLMs can adapt to new tasks with a few input examples in the forward pass. Olsson et al. (2022) hypothesized that ICL is driven by attention heads ("induction heads"), while Xie et al. (2021) interprets ICL as implicit Bayesian Inference process, and Garg et al. (2022) showed that ICL can emerge in the simple case of linear functions. More recently, Hendel et al. (2023) and Todd et al. (2024) hypothesized that ICL creates task (or function) vectors, latent activations that encode the task in LLMs, and Hojel et al. (2024) demonstrated a similar behavior in computer vision models. Huang et al. (2024) proposed to use task vectors in VLMs to compress long prompts that would otherwise not fit in a limited context length. We study how task information evolves within VLMs, specifically the similarity and transferability of the representation when the task is expressed in different modalities.

**Vision-and-Language Models.** Inspired by the success of LLMs, new vision-and-language models (VLMs) have been proposed (Liu et al., 2023a; Li et al., 2023; Tong et al., 2024; Team, 2024; Laurençon et al., 2024; Zhou et al., 2024). Recent VLMs can be roughly categorized to modality late-fusion (Liu et al., 2023a; 2024) and early-fusion (Bavishi et al., 2023; Lu et al., 2022; 2023; Team, 2024) approaches. Late-fusion approaches typically combine a pre-trained visual encoder and LLM by training adapters, potentially with a short end-to-end fine-tuning stage. In contrast, early-fusion approaches focus on end-to-end training without any pre-initialization of the representations. We observe cross-modal task representations for both model categories, suggesting that this property can emerge regardless of the initialization. Several works examine image ICL in VLMs, proposing new models designed for ICL (Alayrac et al., 2022; Laurençon et al., 2024; Doveh et al., 2024; Jiang et al., 2024) and analyzing the impact of in-context example selection on performance (Baldassini et al., 2024). Our work offers a new perspective on image ICL by comparing it with text ICL and demonstrating the similarity between the two processes. We even show VLMs that lack image ICL capabilities (Liu et al., 2023a; Lin et al., 2023; Doveh et al., 2024) can still benefit from task vectors.

## 5    LIMITATIONS

In this work, we demonstrate that VLMs learn cross-modal task representations but we lack a definitive explanation for *why*. Empirical studies offer several hypotheses, such as the existence of isomorphic structures between language and other perceptual representation spaces (Abdou et al., 2021; Patel & Pavlick, 2022; Pavlick, 2023), or representational convergence from modeling the same underlying reality (Huh et al., 2024). Additionally, we observe quantitative improvements for text-to-image transfer but not image-to-text transfer, possibly because VLM training is more text-centric. However, we believe that learning task representations from visual data has its advantages, and we provide qualitative examples where image-to-text transfer proves beneficial.

## 6    CONCLUSION

Vision-and-language models (VLMs) are generalist models capable of solving a wide range of computer vision tasks by framing them as question answering problems in text. Despite their success, we lack a clear understanding of how they work. Our primary observation is that VLMs map inputs into a shared task representation space, regardless of whether the task is defined by text examples, image examples, or explicit instructions. Based on this, we show it is possible to transfer task vectors from one modality (e.g., text) to another (e.g., images). We hope our work will inspire further exploration into the inductive biases of VLMs and the reasons behind their success.

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
