# A    APPENDIX

## A.1    EXPERIMENTAL DETAILS

**Models.** We provide further details on the models used in our evaluation in Table 5.

Table 5: We study a diverse set of representative VLMs spanning both early-fusion and late-fusion paradigms and varying image ICL capabilities.

|  | LLaVA-v1.5 (Liu et al., 2023a) | Mantis-Fuyu (Jiang et al., 2024) | Idefics2 (Laurençon et al., 2024) |
|---|---|---|---|
| Text Model | Vicuna (Chiang et al., 2023) | Fuyu (Bavishi et al., 2023) | Mistral (Jiang et al., 2023) |
| Vision Model | CLIP (Radford et al., 2019) | Fuyu (Bavishi et al., 2023) | SigLIP (Zhai et al., 2023) |
| Paradigm | Late-Fusion | Early-Fusion | Late-Fusion |
| Image ICL | No | Yes | Yes |
| Parameters | 7B | 8B | 8B |
| Num Layers | 32 | 36 | 32 |

**Tasks.** We also show representative examples in Table 1. We scrape the images for all tasks from Wikipedia, because we find that the images tend to depict more clearly identifiable prototypes, unlike traditional computer vision datasets. For some tasks the labels were automatically generated by Claude 3.5 Sonnet (Anthropic, 2024) and manually cross-checked, unless otherwise noted.

- **Country-Capital**. Given the name of the country or its flag, predict the capital city. The text-only case is identical to Todd et al. (2024).
- **Country-Currency**. Given the name of the country or its flag, predict the official currency. The text-only case is almost identical to Todd et al. (2024), except we remove the country modifier from the currency to make the task harder.
- **Animal-Latin**. Given the name of the animal or its image, predict its scientific name in Latin. The labels are derived from the mammals categorized in iNaturalist (iNaturalist, 2017).
- **Animal-Young**. Given the name of the animal or its image, predict the term for its baby.
- **Food-Color**. Given the name of a fruit or vegetable or its image, predict its iconic color. This task is inspired by the conceptual example first proposed in Hendel et al. (2023).
- **Food-Flavor**. Given the name of a fruit or vegetable or its image, predict its iconic flavor profile.

## A.2    EXTENDED EVALUATION TASKS

In our main experiments, we evaluate on six constructed tasks designed to mirror the task types and format proposed by prior work (Hendel et al., 2023; Todd et al., 2024). Here, we automatically construct an "in-the-wild" evaluation set derived from the validation set of VQAv2 (Goyal et al., 2017), which consists of images paired with questions and brief human-annotated answers. To pair each image input with a textual analog, we use dense text descriptions generated by LLaVA-NeXT-34B from the LLaVA-ReCap dataset (Li et al., 2024). Since the dataset includes multiple answers for the same question, we set the ground-truth to be the majority answer. While theoretically only the inputs and answers are needed to construct ICL examples, one problem is that VQAv2 implicitly contains many different tasks that need to be stratified. To overcome this issue, we use the questions to group samples into tasks. We curate a subset of questions asked across a large number of images, such that we can construct a 30-sample validation and 100-sample test set. We report the results on these questions in Table 6, where we see that cross-modal patching (Text ICL xPatch) results in a 6% improvement over few-shot prompting with text examples (Text ICL xBase) and 17% improvement over few-shot prompting with image examples (Image ICL xBase).

The tasks can be enumerated as follows:

- **Food-Class**. Given the image or description, answers: *What kind of food is this?*
- **Shirt-Color**. Given the image or description, answers: *What color is the man's shirt?*

- **Man-Holding**. Given the image or description, answers: *What is the man holding?*

Table 6: We show the test accuracy of cross-modal transfer on image queries for visual question answering tasks derived from VQAv2 (Goyal et al., 2017).

| Model | Food-Class | Shirt-Color | Man-Holding | Avg. |
|---|---|---|---|---|
| **Idefics2** | | | | |
| No Context | 0.00 | 0.00 | 0.00 | 0.00 |
| Image ICL Base | 0.70 | 0.41 | 0.46 | 0.52 |
| Image ICL Patch | 0.49 | 0.19 | 0.39 | 0.36 |
| Text ICL xBase | 0.85 | 0.48 | 0.56 | 0.63 |
| Text ICL xPatch | 0.93 | 0.56 | 0.59 | **0.69** |

## A.3 EXTENDED DISCUSSION OF TEXT ICL TRANSFER

**Evaluating Qwen-VL.** In our main experiments we evaluate on a representative set of early- and late-fusion VLMs enumerated in Table 5. Here, we further verify whether an additional model, Qwen-VL (Bai et al., 2023b), also contains cross-modal task vectors. Similar to LLaVA-v1.5, Qwen-VL is a late-fusion model that fine-tunes a projection from OpenCLIP visual features (Ilharco et al., 2021) into the representation space of the LLM Qwen-7B (Bai et al., 2023a). In Table 7, we report the cross-modal transfer performance for Qwen-VL from text ICL to image queries. Consistent with the trends we observe for both early and late-fusion models Table 3, cross-modal patching (Text ICL xPatch) yields a 22% accuracy improvement over few-shot prompting (Text ICL xBase) across our six cross-modal tasks. Hence, we confirm that cross-modal transfer also benefits Qwen-VL.

Table 7: We show the test accuracy for Qwen-VL when transferring from text ICL to image queries.

| Model | Country-Capital | Country-Currency | Animal-Latin | Animal-Young | Food-Color | Food-Flavor | Avg. |
|---|---|---|---|---|---|---|---|
| **Qwen-VL** | | | | | | | |
| No Context | 0.07 | 0.02 | 0.05 | 0.00 | 0.01 | 0.00 | 0.03 |
| Text ICL xBase | 0.25 | 0.06 | 0.16 | 0.01 | 0.15 | 0.03 | 0.11 |
| Text ICL xPatch | 0.62 | 0.23 | 0.47 | 0.11 | 0.56 | 0.02 | **0.33** |

**Template Format.** While in our main experiments we use the generic template proposed by Todd et al. (2024), here we ablate the usage of a model-specific template for Idefics2. Specifically, we use the recommended template:

$$\texttt{User:}\{x_1\}\texttt{<end\_of\_utterance>}\texttt{\textbackslash nAssistant:}\{y_1\}$$

where we replace the query-answer signifiers (Q, A) with (`User`, `Assistant`), add the special `<end_of_utterance>` token, and delineate each example with `\n\n`. As seen in Table 8, the trends in performance remain consistent with Table 3 – patching cross-modal task vectors significantly outperforms few-shot prompting with text examples.

Table 8: We ablate the template format and display the test accuracy when transferring from text ICL to image queries. We use the recommended template for Idefics2.

| Model | Country-Capital | Country-Currency | Animal-Latin | Animal-Young | Food-Color | Food-Flavor | Avg. |
|---|---|---|---|---|---|---|---|
| **Idefics2** | | | | | | | |
| No Context | 0.00 | 0.00 | 0.07 | 0.00 | 0.00 | 0.00 | 0.01 |
| Text ICL xBase | 0.16 | 0.06 | 0.24 | 0.16 | 0.17 | 0.12 | 0.15 |
| Text ICL xPatch | 0.70 | 0.44 | 0.50 | 0.64 | 0.54 | 0.40 | **0.54** |

**LLM to VLM Transfer.** In Table 9, we display an extended table corresponding to Table 4 in the main text containing the performance when transferring task vectors from the LLM to the VLM.

**Validation Performance.** In our main experiments, we present the test performance of a single model layer, as identified by its average performance across all tasks on the validation set. In Figure 11 we show the performance of all model layers on this validation set. For the late-fusion models, the best task vector lies near the exact middle of the network (Layer 15 / 32 for LLaVA-v1.5 and Layer 16 / 32 for Idefics2). In contrast, for the early-fusion model, the best task vector lies in the late-middle layers (Layer 23 / 36 for Mantis-Fuyu). When comparing tasks, the shape of the curve tends to fall into two categories: a peak then plateau (Food-Color, Food-Flavor) or single sharp

Table 9: We show the test accuracy when transferring task vectors from text ICL in the LLM to image queries in the VLM.

| Model | Country-Capital | Country-Currency | Animal-Latin | Animal-Young | Food-Color | Food-Flavor | Avg. |
|---|---|---|---|---|---|---|---|
| **LLaVA-v1.5** | | | | | | | |
| VLM-VLM xPatch | 0.31 | 0.30 | 0.26 | 0.18 | 0.53 | 0.31 | 0.32 |
| LLM-VLM xPatch | 0.33 | 0.32 | 0.25 | 0.33 | 0.53 | 0.45 | **0.37** |
| **Idefics2** | | | | | | | |
| VLM-VLM xPatch | 0.61 | 0.40 | 0.48 | 0.62 | 0.53 | 0.39 | 0.51 |
| LLM-VLM xPatch | 0.57 | 0.58 | 0.46 | 0.55 | 0.54 | 0.39 | **0.52** |

peak (all other tasks). We hypothesize that the shape is associated with the diversity of the output space – fewer possible outputs make it more likely for later layers, which are closer to the answer representation, to yield a plausible result.

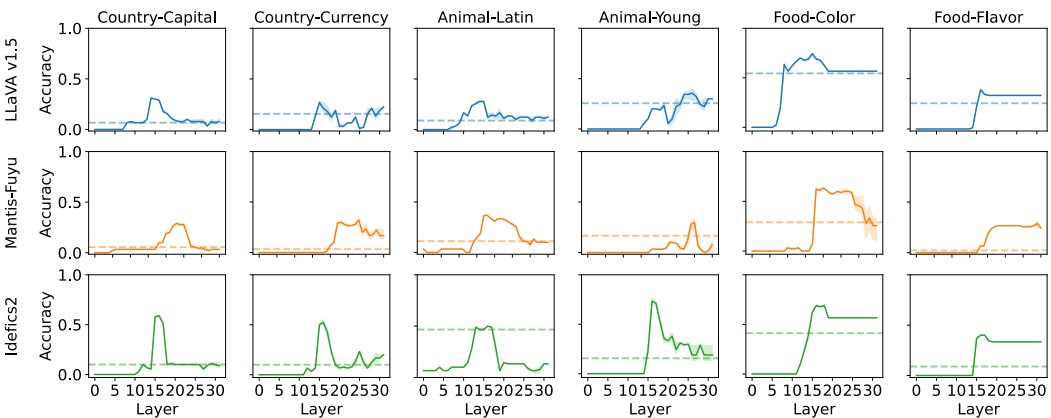

Figure 11: We display validation performance for transferring task vectors from text ICL to image queries (xPatch) across model-task combinations. Each subplot shows the accuracy by model layer, with a dotted line providing the Text ICL baseline (xBase) accuracy for reference.

### A.4 Ablating All Modality Combinations

In Table 10, we display additional results when patching task vectors in all combinations of example-query modality. For image queries, the cross-modal setting is highly beneficial, where task vectors derived from text ICL outperform those from image ICL by 11-20% respectively. For text queries, this is not the case, where the cross-modal setting underperforms by 9-23%. In Sec. 3.3 we discuss the challenges in benchmarking transfer from image ICL examples to text queries. We think that an evaluation suite for identifying visual concepts from dense text descriptions would benefit more from cross-modal transfer, which is an exciting area of further research.

Table 10: We display the test accuracy when patching task vectors in all combinations of example-query modality. The best-performing combination for a given query modality is highlighted. Each setting is denoted as {ICL Modality}-{Query Modality}. The best-performing combination for a given query modality is highlighted.

| Model | Country-Capital | Country-Currency | Animal-Latin | Animal-Young | Food-Color | Food-Flavor | Avg. |
|---|---|---|---|---|---|---|---|
| **LLaVA-v1.5** | | | | | | | |
| Image - Image Patch | - | - | - | - | - | - | - |
| Text - Image xPatch | 0.31 | 0.30 | 0.26 | 0.18 | 0.53 | 0.31 | 0.32 |
| Text - Text Patch | 0.97 | 0.58 | 0.77 | 0.20 | 0.63 | 0.41 | 0.59 |
| Image - Text xPatch | - | - | - | - | - | - | - |
| **Mantis-Fuyu** | | | | | | | |
| Image - Image Patch | 0.17 | 0.03 | 0.16 | 0.05 | 0.50 | 0.31 | 0.20 |
| Text - Image xPatch | 0.32 | 0.23 | 0.36 | 0.09 | 0.51 | 0.36 | 0.31 |
| Text - Text Patch | 0.46 | 0.30 | 0.48 | 0.18 | 0.28 | 0.36 | 0.34 |
| Image - Text xPatch | 0.31 | 0.01 | 0.36 | 0.05 | 0.40 | 0.34 | 0.25 |
| **Idefics2** | | | | | | | |
| Image - Image Patch | 0.58 | 0.32 | 0.40 | 0.03 | 0.39 | 0.17 | 0.31 |
| Text - Image xPatch | 0.61 | 0.40 | 0.48 | 0.62 | 0.53 | 0.39 | 0.51 |
| Text - Text Patch | 0.97 | 0.61 | 0.74 | 0.54 | 0.63 | 0.41 | 0.65 |
| Image - Text xPatch | 0.81 | 0.43 | 0.58 | 0.04 | 0.40 | 0.27 | 0.42 |

## A.5 EXTENDED DISCUSSION OF INSTRUCTION TRANSFER

**Quantifying Task Conflict.** In Figure 9, we present examples where the model encounters conflicting tasks, representing a practical scenario in which the user's prompt clashes with the global system instruction. Here, we conduct an extended analysis to quantify the effectiveness of cross-modal patching for enforcing this textual system instruction on image queries. First, we select 100 random pairs of conflicting questions for the same image, derived from the validation set of VQAv2 (Goyal et al., 2017). One question is designated as the "conflicting task," where we measure the rate at which the model is able to produce the majority-annotated answer for this question. As seen in Table 11, patching instruction vectors (Instruction xPatch) is highly effective in steering the model toward a different task, outperforming the common alternative of including the instruction in the system prompt (System Prompt) by 27%. Additionally, we include a baseline (Instruction xBase) where the model is not provided with the new instruction.

Table 11: Instruction xPatch effectively steers the model to perform a newly introduced (and conflicting) task.

| Method | Acc. |
|---|---|
| Instruction xBase | 0.04 |
| Instruction xBase + System Prompt | 0.05 |
| Instruction xBase + Instruction xPatch | **0.32** |

## A.6 EXTENDED DISCUSSION OF IMAGE ICL TRANSFER

**Quantifying Image ICL Transfer.** In Figure 10, we illustrate a few cases where image ICL examples can be helpful for text queries, particularly for tasks that involve recognizing visual concepts in dense text descriptions. We conduct a small-scale analysis on the 12 samples presented, where for each sample we curate corresponding images and corresponding dense text descriptions for the input and a pre-defined ground truth answer for the output. Each sample is used as a held-out query, while the remaining samples are used as $N = 3$ ICL examples. For dense text descriptions as the query, we compare the performance of the cross-modal image ICL and unimodal text ICL examples. As seen in Table 12, cross-modal image ICL examples are much more effective when patched rather than few-shot prompted, where Image ICL xPatch outperforms Image ICL xBase by 17%. Cross-modal patching is also competitive with the unimodal baselines, where Image ICL xPatch improves over both Text ICL xBase and Text ICL xPatch by 8%. Hence, we see that concept recognition in dense text descriptions is a promising area in which image ICL can be useful, and we think it is a promising direction for future research and larger-scale evaluation.

**Dense Text Descriptions.** Corresponding to Figure 10, we display the text descriptions used in text ICL designed to be analogous with the images used in image ICL.

- {*The logo is a rainbow-colored apple.* : **Apple**}

Table 12: Accuracy of transfer from image ICL to dense text descriptions.

| Model | Acc. |
|---|---|
| No Context | 0.00 |
| Text ICL Base | 0.17 |
| Text ICL Patch | 0.17 |
| Image ICL xBase | 0.08 |
| **Image ICL xPatch** | **0.25** |

- {*The logo is a white ghost against a yellow background.* : **Snapchat**}
- {*The logo is a white camera against a gradient background.* : **Instagram**}
- {*The logo is the letter P stylized to look like a pushpin.* : **Pinterest**}
- {*The character is a squirrel wearing an astronaut suit.* : **Sandy Cheeks**}
- {*The character is a puffer fish wearing a blue shirt, red skirt, and blue hat.* : **Mrs. Puff**}
- {*The character is a crab wearing a blue shirt, blue pants, and brown belt.* : **Mr. Krabs**}
- {*The character is a pink starfish wearing green and purple pants.* : **Patrick Star**}
- {*An image of an orange and white cat wearing a blue shirt playing the keyboard.* : **Keyboard Cat**}
- {*An image of a shiba inu sitting on a couch.* : **Doge**}
- {*A cartoon of a dog wearing a hat sitting in a room engulfed with flames.* : **This Is Fine Dog**}
- {*An image of an unhappy cat with blue eyes and white and brown fur.* : **Grumpy Cat**}

## A.7 TOKEN REPRESENTATION EVOLUTION FOR ALL TASKS

**Implementation Details.** For this experiment, we condition the model on some task specification (e.g., text ICL, image ICL) and cache the intermediate activation across all model layers. Following logit lens (nostalgebraist, 2020), we normalize and project the activation with the model's unembedding matrix, which produces a probability distribution over all vocabulary tokens. We aggregate statistics over 100 activations produced by different runs specifying the same task. In our discrete visualization (Figure 4), we collect the top-1 token with the highest probability across runs and visualize the tokens in a pie chart. In our continuous visualization (Figure 3), we compare the relative probability of three pre-defined tokens corresponding to the input, task, and answer. Specifically, we use the token *auf* for the input, one of {*capital, currency, species, baby, color, flavor*} for the task, and each run's ground-truth label for the answer. For each run, we take the softmax of the three token probabilities to obtain a normalized probability distribution. We plot the mean and variance across runs for these token probabilities as a line chart.

**Discrete Visualization.** We show an expanded series of pie charts depicting the representation evolution for all tasks in Figure 12, corresponding to Figure 4 of the main text.

**Continuous Visualization.** We provide an expanded series of line graphs showing the representation evolution for all tasks in Figure 13, corresponding to Figure 3 of the main text.

**Conditioning on Instructions.** We visualize the token representation evolution when conditioning on instructions rather than examples in Figure 14 and Figure 13. We do not display discrete pie charts since a single instruction does not produce aggregate statistics, unlike examples where there are multiple possible sets. The instruction-based vector decodings are often interpretable and resemble a meta summary for the task, similar to the observations in Sec. 2.3.

**t-SNE Visualization.** In Figure 15, we compare task vectors defined with different specification methods (Text ICL, Image ICL, and Instruction) by visualizing them in the same embedding space via t-SNE (van der Maaten & Hinton, 2008). The ideal cross-modal representation space would display clusters with distinct colors (denoting different tasks) composed of intermixed shapes (denoting different specifications). At first, each setting is in its own distinct cluster, where different specifications for the same task are clearly separated. Then, the clusters for these different specifications move closer together until they finally mix fully. While most tasks (green, red, blue, orange) exhibit the ideal clustering, the food-related tasks (purple, brown) do not. We hypothesize that this is the case because the color and flavor of a food are fairly correlated, resulting in the lack of separation between the tasks.

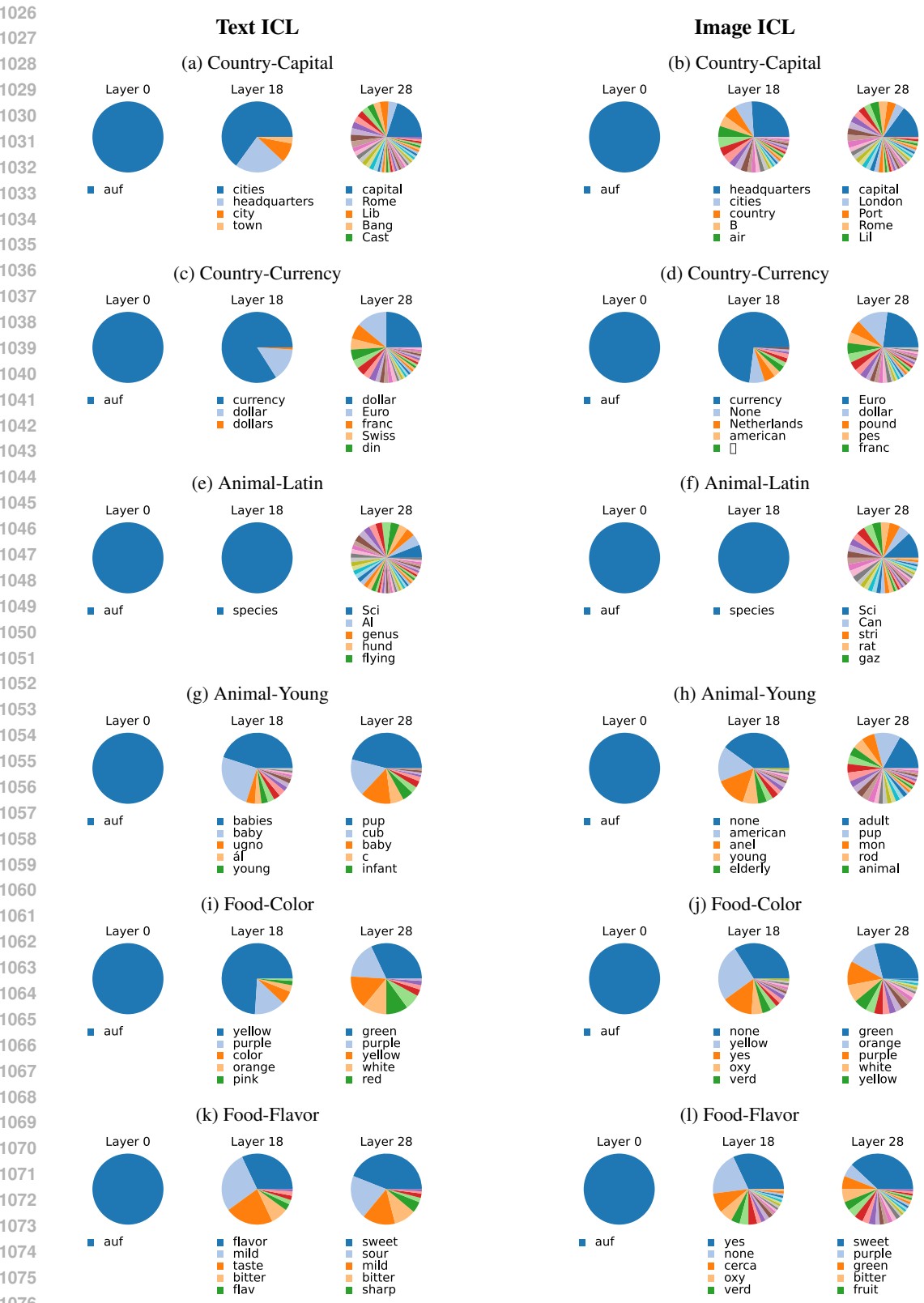

Figure 12: We show a discrete visualization of how the token representation evolves across layers for all tasks. Each pie chart slice represents a top-1 decoding across 100 sets of examples, and the most common decodings are displayed below.

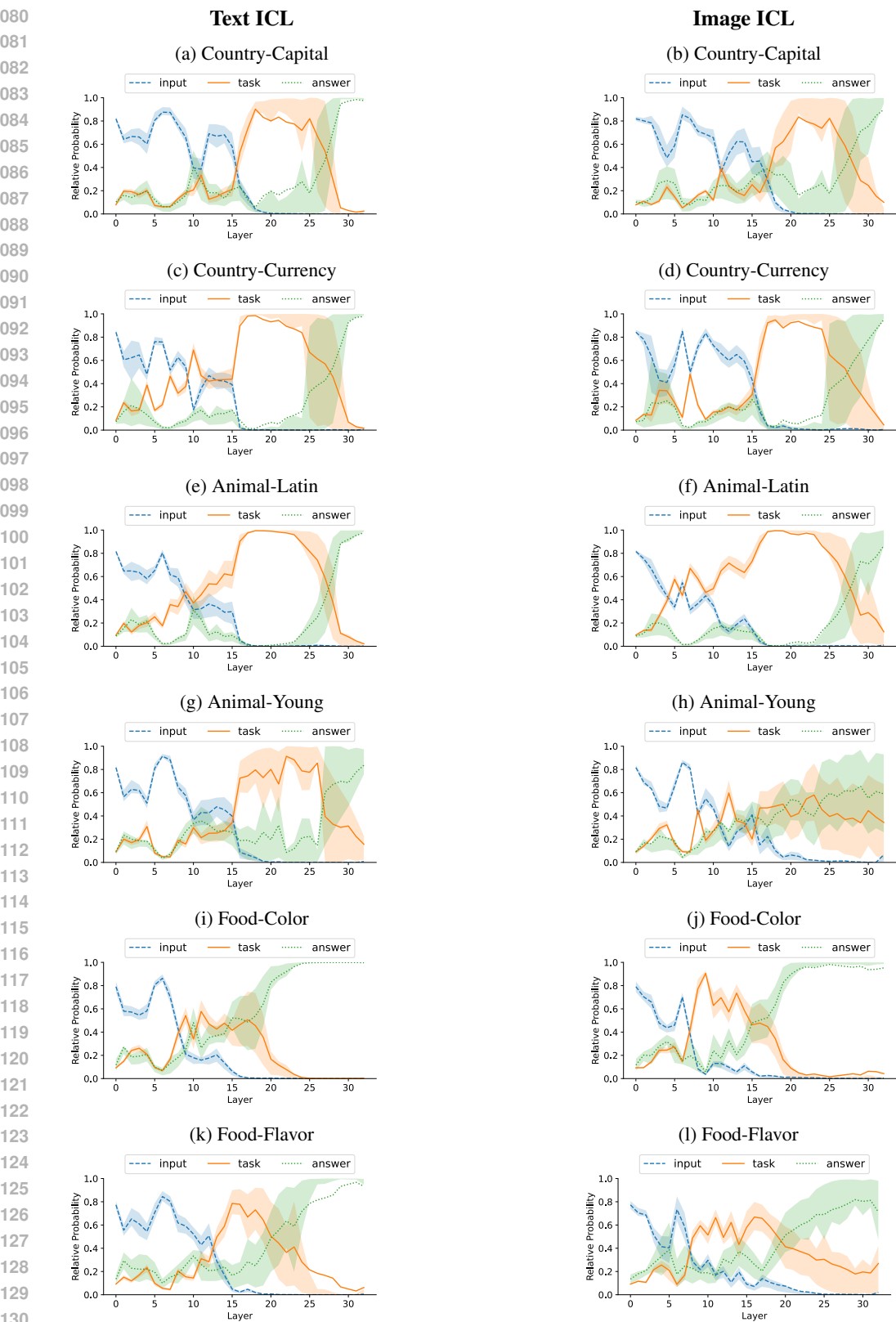

Figure 13: We show a continuous visualization of how the token representation evolves across layers for all tasks. Each line shows the representational similarity with a pre-defined token, aggregated over 100 sets of examples. We use the token *auf* for the input, one of {*capital, currency, species, baby, color, flavor*} for the task, and each run's ground-truth label for the answer.

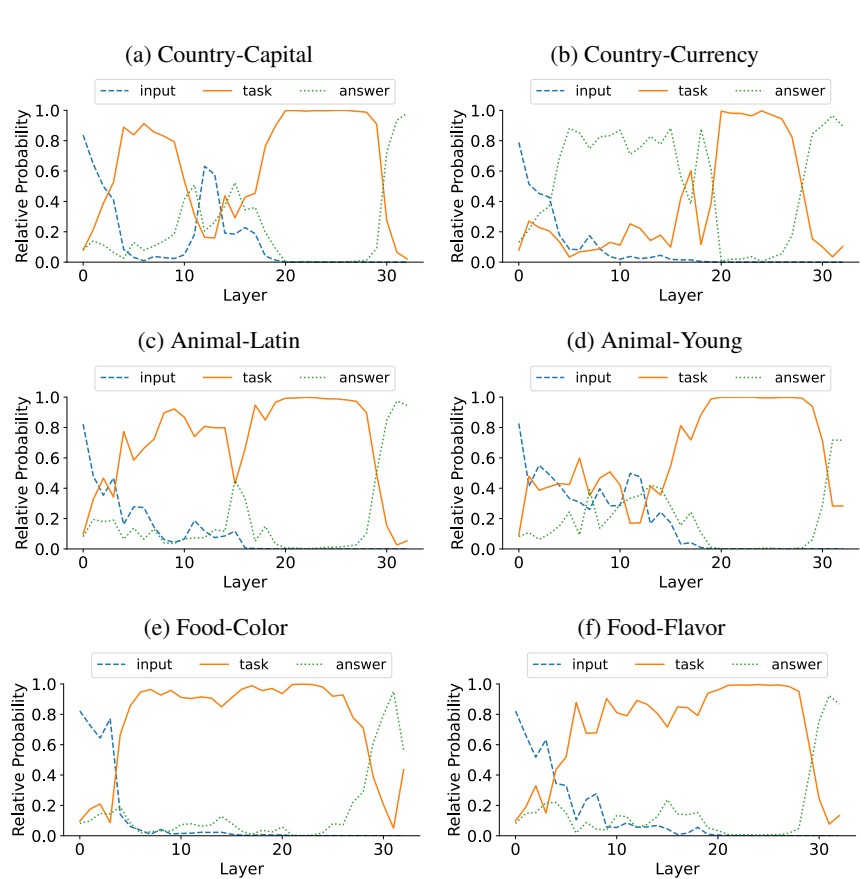

Figure 14: We show a continuous visualization of the token representation evolution when conditioned on instructions rather than examples. The results are aggregated over a single instruction rather than multiple examples, so there are no variance bars.

Table 13: We depict the top-5 decodings for the instruction-based vector, where ◊ denotes symbols that do not correspond to common word tokens.

| Task | Instruction |
|---|---|
| Country-Capital | *city*, *GU*, *vik*, *cities*, *headquarters* |
| Country-Currency | ◊, ◊, ◊, *itos*, ◊ |
| Animal-Latin | *species*, *genus*, ◊, *animals*, *american* |
| Animal-Young | *baby*, *babies*, ◊, *bach*, *called* |
| Food-Color | *colors*, *color*, *colour*, *ETH*, *ilo* |
| Food-Flavor | *taste*, *tastes*, *arom*, *food*, *flavor* |

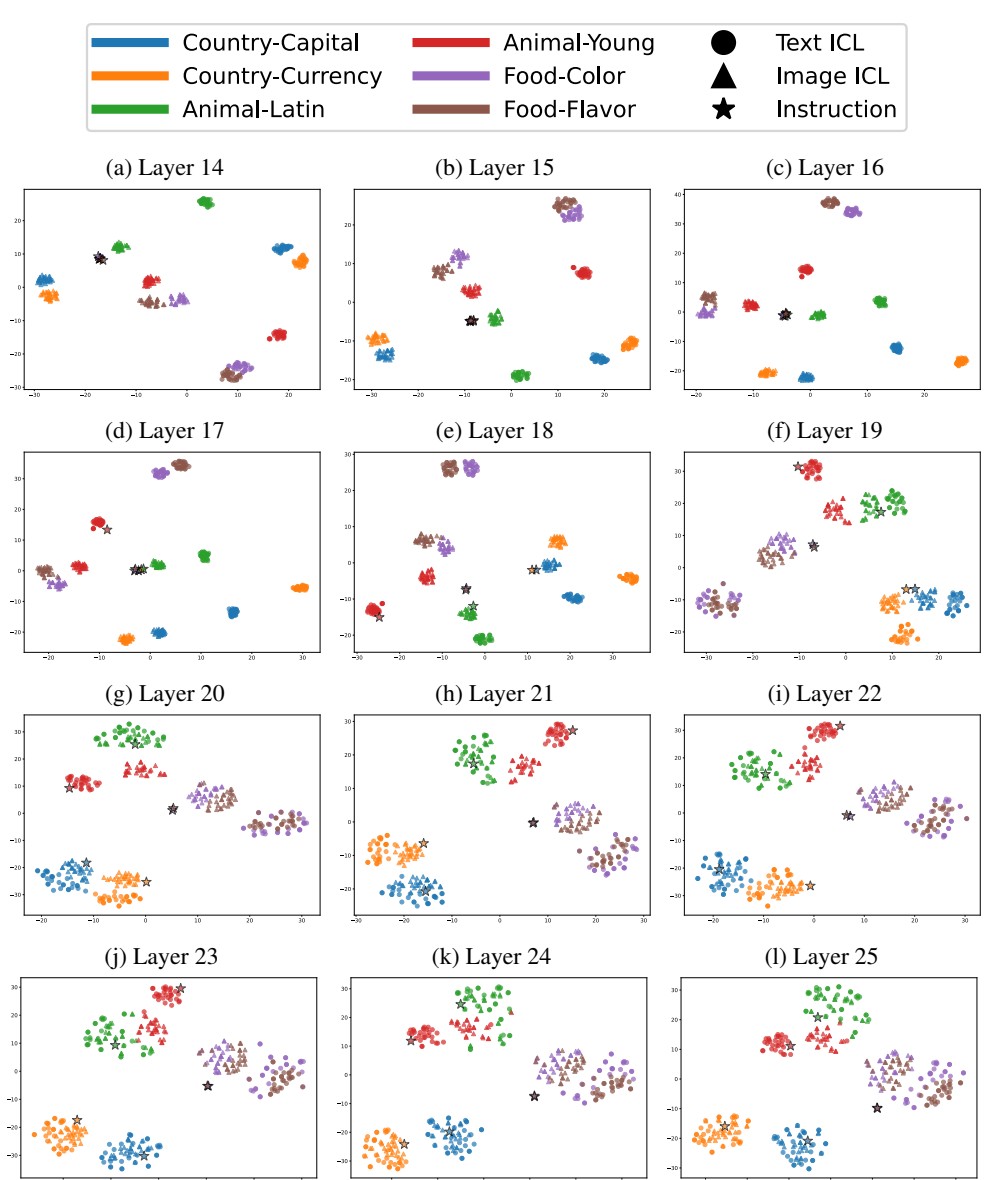

Figure 15: We use t-SNE (van der Maaten & Hinton, 2008) to visualize the embedding space of task vectors for different tasks (denoted by color) defined with different specification methods (denoted by shape) across model layers. Each point represents a set of text or image ICL examples, or a single instruction.