# OpenReview forum: "Task Vectors are Cross-Modal"
_ICLR.cc/2025/Conference — Submitted to ICLR 2025_

### Official Review · Reviewer_23Jk · 2024-11-03

**Soundness:** 3
**Presentation:** 3
**Contribution:** 3
**Rating:** 5
**Confidence:** 3

**Summary:**

The paper investigates the internal representations of vision-and-language models , specifically focusing on how task vectors encode information across different modalities. This paper present findings that conceptually similar tasks, whether specified through examples or instructions, yield similar vector representations in VLMs. This property enables task vectors to be transferred from one modality to another (e.g., from text to image), with the ensembling of exemplar and instruction-based vectors producing superior task representations. The research contributes to understanding the underlying mechanisms of VLMs, emphasizing their ability to maintain consistent task representations across modalities.

**Strengths:**

1. The paper introduces cross-modal task vectors, demonstrating that VLMs can generalize tasks across different input modalities (e.g., text to image) effectively,advancing multi-modal interpretability.
2. The paper reveals a consistent three-stage token evolution (input, task, answer) across modalities, providing deeper understanding of VLM internal mechanisms.
3. The finding that task vectors from language-only models can be applied in VLMs underscores the versatility of cross-modal task representations.

**Weaknesses:**

1. **Insufficient Validation Across Task Types**: The paper's validation is limited to a narrow range of task types. Expanding the evaluation to diverse tasks would strengthen its claims.

2. **Unclear description of key elements**: The method for obtaining task vectors is not described clearly, particularly regarding the specific settings and conditions.It would be beneficial to have a clearer explanation of the context and conditions under which these task vectors are derived, including any specific parameters or preprocessing steps involved.

3. **Lack of Methodological Details**: Key methodological details are not adequately explained, particularly in terms of the implementation of xPatch and xBase. The paper would benefit from a more in-depth description of how these configurations are set up and executed. Additionally, the combination of Instruction and Exemplar vectors in the Instruction + Exemplar xPatch approach is not described in sufficient detail. A clearer explanation of how these vectors are integrated would greatly enhance the paper’s clarity and reproducibility.

**Questions:**

See the section on weakness. We will increase the score based on the answer to the question.

---

> ### Author Response · Authors · 2024-11-22
>
> Thank you for your helpful review; we have addressed all of the listed weaknesses and revised the manuscript according to your suggestions. We hope that these answers will encourage you to increase your score.
>
> ---
>
> 1. *Insufficient Validation Across Task Types.*
>
> Thank you for your suggestion; **we have added an evaluation on three additional visual question answering tasks derived from VQAv2 [1] in Sec A.2 of the Appendix.** We copy the results to Table 1 for reference, where we show that **cross-modal patching results in a 6% improvement over few-shot prompting with text examples (Text ICL xPatch vs. Text ICL xBase) and 17% improvement over few-shot prompting with image examples (Text ICL xPatch vs. Image ICL xBase).**
>
> **Table 1. We show the test accuracy of cross-modal transfer on image queries for visual question answering tasks derived from VQAv2.**
> | **Model**         | **Food-Class** | **Shirt-Color** | **Man-Holding** | **Avg.** |
> |--------------------|----------------|-----------------|-----------------|----------|
> | No Context       | 0.00           | 0.00           | 0.00           | 0.00     |
> | Image ICL Base   | 0.70           | 0.41           | 0.46           | 0.52     |
> | Image ICL Patch    | 0.49           | 0.19           | 0.39           | 0.36     |
> | Text ICL xBase   | 0.85           | 0.48           | 0.56           | 0.63     |
> | **Text ICL xPatch** | __0.93__       | __0.56__       | __0.59__       | **0.69** |
>
>
> [1] Goyal et. al. Making the V in VQA Matter: Elevating the Role of Image Understanding in Visual Question Answering. CVPR 2017.
>
> ---
>
> 2. *Unclear description of key elements.*
>
> *Method for obtaining task vectors.* **We have revised Sec 2.2 in the main text to more thoroughly explain our method for obtaining task vectors. We have also updated [Figure 2a](https://github.com/t4345254/anon_iclr/blob/main/approach_detailed.pdf) to include a more detailed illustration.** As stated in L143, to extract task vectors, we do the following. We run two forward passes: one to extract the task vector from the text ICL examples and one with a contextless image query. We extract the task vector the $l$-th transformer layer output at the delimiter token between the last input-output pair $(x_N, y_N)$, and we inject it directly at the corresponding layer and token position of the query. Our only hyperparameter is the best layer to patch, which we determine via the average task accuracy on the validation set.
>
> *Specific parameters or preprocessing steps.* **We also discuss the context and conditions under which these task vectors are derived in L314 of the main text.** When specifying a task via exemplars, we use $N=5 $ ICL examples. We also preprocess images such that they are a standard width of 224 pixels.
>
> ---
> 3. *Lack of Methodological Details.*
>
>
> **In [Figure 2](https://github.com/t4345254/anon_iclr/blob/main/approach_detailed.pdf) of the main text, we have added an additional visualization explaining our configurations, including the implementation of xPatch and xBase as well as Exemplar + Instruction xPatch.**
>
> *Implementation of xPatch and xBase.* Please see response #2 above for a summary regarding xPatch. The setting xBase denotes the few-shot prompting baseline, where the ICL examples and query are jointly fed to the transformer in the same prompt.
>
> *Combination of Instruction and Exemplar vectors.* For Exemplar + Instruction xPatch, we feed a random set of ICL examples in one forward pass to obtain the exemplar-based task vector as well as an instruction specifying the same task in another forward pass to obtain the instruction-based vector. To integrate the vectors, we compute their average, or element-wise mean. We then patch this ensembled vector onto an unseen image query, which we evaluate in Figure 8 of the main text.

---

> > ### Author Response · Authors · 2024-11-25
> > **Response to Reviewer 23Jk**
> >
> > We are bumping this thread, since we have not yet received a response from Reviewer 23Jk. We would like to highlight for Reviewer 23Jk that we have added the suggested additional tasks and addressed the listed weaknesses in our revisions to the manuscript. **Specifically, we have added Sec. A.2 to the Appendix and updated Sec. 2.2 of the main paper, which has improved the paper with the help of your feedback.** We invite Reviewer 23Jk to review these updates and consider increasing your score if your concerns have been adequately addressed.

---

### Official Review · Reviewer_x3D1 · 2024-11-03

**Soundness:** 2
**Presentation:** 1
**Contribution:** 3
**Rating:** 3
**Confidence:** 3

**Summary:**

This paper presents a method for identifying task vectors in the residual space of a auto-regressive VLM. Using a few samples of a task in one modality, the model can be patched to perform the same task in the other modality.

**Strengths:**

Originality:
 - The paper is the first to explore the area of multi-modal fewshot task learning via patch learning

Quality:
 - The experiments seem to fully explore the task-space in the domain of these llava-like models

Clarity:
 - The figures are intuitive and the results are clearly presented

Significance:
 - I personally think mechanistic interpretibility / model steering will be a new foundational paradigm, so I like to see more work in this space.

**Weaknesses:**

The main issue with the paper is it's clarity. It is incredibly difficult to read and follow. I did not realize the proposed method was for auto-regressive VLMs like LLava until page 5! The notation used throughout the paper is very unusual and led me to believe this method was for CLIP-like models. It wasn't until I did an external literature review were I found the prior work "Find Visual Task Vectors" uses the identical notation. While I'm not against maintaining the notation, it's clear some of the equations are for image generative models, not auto-regressive text generative models.

Patching itself is never actually described, not clear how the proposed method actually works at test time. Are there any hyper-parameters used? I know when I run model steering vectors, I'll normalize the vector and add it by 3x, is that happening here?

Logit lens is introduced and used without any explanation as to how it works. I have no idea how the experiment in figure 3 works.

I'm also disappointed mechanistic interpretability / model steering isn't put more front-in-center, as the proposed method is an application of model steering/patching (see Neel Nanda's cited papers). I felt a little bit misdirected.

Overall, the writing is a major hindrance to my judgement for the other aspects of the paper. I currently do not feel qualified to evaluate the experiments given their current descriptions. As a single example, I need more explanation of what "Exemplar xPatch" is and "Instruction + Exemplar xPatch" is. But to be clear, I have similar clarity issues for all the experiments.

While I believe the proposed method would be of interest to the community, I cannot recommend the paper in its current form.

Minor:
 - table 1 has a hard time rendering on my computer. Can you lower the resolution quality of the images in it? I think the strawberry is extremely high resolution.

**Questions:**

See weaknesses.

---

> ### Author Response · Authors · 2024-11-22
>
> Thank you for your detailed comments on the writing of the paper. We have incorporated all the suggested revisions to enhance the clarity of our method and experiments. We hope these improvements address your concerns and kindly ask you to reconsider your score.
>
> ---
>
> 1. *I did not realize the proposed method was for auto-regressive VLMs.*
>
> Thank you for pointing out the ambiguity of the term “VLM.” **We have revised Figure 1, the abstract, and the first paragraph of the introduction to make it more explicit that we study autoregressive VLMs.**
>
> **Following your comment about the notation, we have revised Sec. 2.1 and Sec. 2.2 of the main text to emphasize that we study autoregressive models.** Additionally, in L131 of the main text we make it clear that “For autoregressive models, i.e., the LLMs studied in prior work and the VLMs we study, [the forward pass] represents a distribution for the next token prediction.”
>
> ---
>
> 2. *Patching itself is never actually described.*
>
> **We have revised Sec 2.2 in the main text to more clearly describe the patching process. We have also updated [Figure 2a](https://github.com/t4345254/anon_iclr/blob/main/approach_detailed.pdf) to include a more detailed illustration.**
>
> As stated in L143, at test time, the proposed cross modal patching method is implemented as follows. We run two forward passes: one to extract the task vector from the text ICL examples and one with a contextless image query. We extract the task vector the $l$-th transformer layer output at the delimiter token between the last input-output pair $(x_N, y_N)$, and we inject it directly at the corresponding layer and token position of the query.
>
> ---
>
> 3. “Are there any hyper-parameters used? I know when I run model steering vectors, I'll normalize the vector and add it by 3x, is that happening here?”
>
> We have a single hyperparameter, which is the position of the layer we patch into. We determine the best layer for each model based on the average task accuracy on the validation set.
>
> We replace the activation with the raw layer output; we do not do any post-hoc normalization or scaling. This is the same setup as prior work on task vector patching in language models [1].
>
> [1] Hendel et. al. In-Context Learning Creates Task Vectors. Findings of EMNLP 2023.
>
> ---
>
> 4. *Logit lens is [...] used without any explanation as to how it works.*
>
> Thank you for your feedback on Sec 2.3. **We have added an additional section called “Implementation Details” in Sec A.7 of the Appendix, which discusses logit lens and our token representation evolution experiment in more detail.**
>
> As stated in L994, for the experiment in Figure 3 of the main text, we normalize and project the activation with the model’s unembedding matrix (i.e., logit lens), which produces a probability distribution over all vocabulary tokens. We take the probabilities of three pre-defined tokens corresponding to the input, task, and answer, and convert them into relative probabilities by taking the softmax of these three token probabilities. We aggregate these statistics for 100 activations from different runs specifying the same task, plotting the mean and variance as a line chart.

---

> > ### Author Response · Authors · 2024-11-22
> >
> > 5. *[M]echanistic interpretability [...] [could be] put more front-in-center.*
> >
> > Thank you for the comment. Indeed, we view our work as part of the broader mechanistic interpretability field. **To better emphasize this, we added the following lines to Sec. 4 of the main paper, the Related Work, which discusses mechanistic interpretability already: “Here, we use Activation Patching to demonstrate that task representations transfer across modalities, regardless of being specified by examples or instructions.”** We are also open to moving the Related Work to the front of the paper if you think this would help better position the paper.
> >
> > ---
> >
> > 6. *I currently do not feel qualified to evaluate the experiments given their current descriptions.*
> >
> > **We have revised Sec 2.2 of the main text to more clearly enumerate all configurations and align each experiment with a detailed description. We hope you can reconsider your score in light of these revisions.**
> >
> > We summarize our main experiments below for reference.
> > - **Text ICL Transfer.** First, we evaluate patching text ICL examples onto image queries, which we call *Text ICL xPatch* and evaluate in Table 3. We also look at the special case of transferring task vectors from a base LLM to its fine-tuned VLM, which we call *LLM-VLM xPatch* and is evaluated in Table 4.
> > - **Instruction Transfer.** Next, we consider patching text instructions onto image queries, which we call *Instruction xPatch.* While prior work only studies exemplars, we also consider instructions. We explore the utility of such instructions for making exemplar-based task vectors more robust, which we denote as *Exemplar + Instruction xPatch* and evaluate in Figure 8. We also look at a scenario of conflicting instructions, denoted as *Instruction xBase vs. Instruction xPatch*, as illustrated in Figure 9.
> > - **Image ICL Transfer.** Finally, we also consider patching image ICL examples onto text queries, which we call *Image ICL xPatch.* We show that such transfer can be useful for tasks that map a dense textual description to its underlying visual concept, as shown in Figure 10.
> >
> > ---
> >
> > 7. *I need more explanation of what "Exemplar xPatch" is and "Instruction + Exemplar xPatch" is.*
> >
> > **We have added [Figure 2c](https://github.com/t4345254/anon_iclr/blob/main/approach_detailed.pdf) to the main text to better illustrate Exemplar xPatch and Instruction + Exemplar xPatch.**
> >
> > Please see response #2 above for a more detailed description of Exemplar xPatch. For Exemplar + Instruction xPatch, we feed a random set of ICL examples in one forward pass to obtain the exemplar-based task vector as well as an instruction specifying the same task in another forward pass to obtain the instruction-based vector. We ensemble the vectors by computing their average, or element-wise mean, which we then patch onto an unseen image query.
> >
> > ---
> >
> > 8. *[T]able 1 has a hard time rendering on my computer.*
> >
> > Thank you for surfacing this rendering issue; we have compressed all images in the pdf and updated the manuscript accordingly.

---

> > > ### Author Response · Authors · 2024-11-25
> > > **Response to Reviewer x3D1**
> > >
> > > We are bumping this thread, since we have not yet received a response from Reviewer x3D1. We would like to highlight for Reviewer x3D1 that we have made the suggested revisions to the manuscript to improve the paper’s clarity. **Specifically, we have significantly revised the notation and writing in Sec. 2.1 and Sec. 2.2 of the main paper, added Sec. A.7 to the Appendix, and updated portions of the Abstract, Introduction, and Related Work in the main paper, which has improved the presentation of the paper with the help of your feedback.** We invite Reviewer x3D1 to review these updates and consider increasing your score if your concerns have been adequately addressed.

---

### Official Review · Reviewer_oiXL · 2024-11-04

**Soundness:** 2
**Presentation:** 3
**Contribution:** 2
**Rating:** 3
**Confidence:** 3

**Summary:**

This paper explores how vision-and-language models (VLMs) encode tasks across different modalities. The authors demonstrate that task vectors are cross-modal, enabling task information to transfer between modalities, which boosts performance. They further show that combining text instructions with examples enhances the sample efficiency of task vectors. Experimental results support all these findings.

**Strengths:**

1. This paper is well-written and easy to understand.
2. Discovering cross-modal task vectors and their transferability is an innovative approach.
3. The experiment appears to be thoughtfully designed and well-executed.

**Weaknesses:**

1. The methodology is straightforward. Although the authors mention that task vectors are cross-modal and can transfer between different modalities, this work appears to be an extension of previous work on function vectors. The method seems like a direct extension, which is trivial and lacks novelty.
2. Figure 2 suggests the possibility of transferring text ICL vectors to image-based tasks, but the specifics of the patching process are not thoroughly explained.
3. Mapping the input space to a vector, represented by $G$, is essential for generating task vectors; however, the authors do not clearly explain how this mapping is calculated.
4. The method applies only to MLLM types that use a feature encoder, limiting its generalizability. For example, models like QWen, which lack a feature encoder, would not be compatible with this approach.
5. In lines 241–244, the authors initially state that the top-1 decoding for both text and image ICL are similar, but they then claim that alignment with language is "not immediately obvious" for image ICL. This seems contradictory, as the table does not show significant differences between text and image ICL. Additionally, the explanation that "the model could have mapped the task vectors close to unused nonsense tokens" lacks supporting evidence.
6. For several experimental results, such as task conflict and image ICL transfer, the paper primarily presents qualitative examples as evidence. Providing quantitative results would strengthen the claims.

**Questions:**

1. The paper does not include an Appendix, although some sentences in the main text reference it.
2. Line 351 should refer to Table 4 instead of Figure 4.

**Details Of Ethics Concerns:**

No ethics concerns

---

> ### Author Response · Authors · 2024-11-22
>
> Thank you for your valuable comments and constructive feedback. We have thoroughly revised the manuscript, incorporating all the suggested quantitative results to strengthen our claims. When referencing the Appendix, we are referring to a separate pdf uploaded under “Supplementary Material.” We hope that the below points help clarify the scientific merit of our work and encourage you to reconsider your score.
>
> ---
>
> 1. *The methodology is straightforward.*
>
> We agree that the methodology we use is straightforward. **We see this simplicity as a strength, as it demonstrates that the phenomenon we discover is robust and does not require additional methodological complexity to be observed.** Specifically, VLMs generate task vectors to produce answers, and these task vectors are shared across modalities.
>
> **Additionally, we strongly disagree with the characterization of our work as a trivial extension of past research. We highlight the main differences from Function Vectors [1]:**
>
> Unlike [1], we analyze how tokens evolve from input to output when VLMs generate answers. We demonstrate that ICL behaves similarly whether using text or image representations (see Figures 3, 13). This empirical finding is novel, and the existence of cross-modal task vectors in VLMs is a byproduct of it. Previous works did not analyze how tokens evolve.
>
> Unlike past works, we discover that tasks can be defined by explicit textual instructions and by image examples, whereas past work only explored textual examples.
>
> We are the first to report these findings, which introduces new scientific knowledge of interest to the community. We think it is unfair to discount these contributions due to the absence of unnecessary methodological complexity, and **we hope you can reconsider your final score in light of our novel findings.**
>
> [1] Todd et. al. Function Vectors in Large Language Models. ICLR 2024.
>
> ---
>
> 2. *[T]he patching process [is] not thoroughly explained.*
>
> **We have revised Sec 2.2 in the main text to more thoroughly explain the patching process. We have also updated [Figure 2a](https://github.com/t4345254/anon_iclr/blob/main/approach_detailed.pdf) to include a more detailed illustration.**
>
> As stated in L143, to calculate the mapping from the input space to a vector, we do the following. We run two forward passes: one to extract the task vector from the text ICL examples and one with a contextless image query. We extract the task vector the $l$-th transformer layer output at the delimiter token between the last input-output pair $(x_N, y_N)$, and we inject it directly at the corresponding layer and token position of the query.  Unlike prior work which only looks at textual exemplars, we also explore extracting task vectors from instructions and image exemplars.
>
> ---
>
> 3. *The method applies only to MLLM types that use a feature encoder.*
>
> **We actually do show that our method applies to MLLMs without a feature encoder like Mantis-Fuyu**, see the original submission (Table 3 of the main text). To quote the description of the model from [2], Mantis-Fuyu is a model that has “no specialized image encoder” where “[image] patches are linearly projected directly into the first layer of the transformer.”
>
> Nevertheless, we followed your suggestion and added an evaluation of Qwen-VL [3]. **We show that for Qwen-VL, cross-modal patching yields a 22\% accuracy improvement over few-shot prompting across our six cross-modal tasks  (Text ICL xPatch vs. Text ICL xBase) .**  We include this table in Sec. A.3 of the Appendix.
>
> **Table 1. Test accuracy of Qwen-VL when transferring from text ICL to image queries.**
> | Model             | Country-Capital | Country-Currency | Animal-Latin | Animal-Young | Food-Color | Food-Flavor | Avg.  |
> |-------------------|-----------------|------------------|--------------|--------------|------------|-------------|-------|
> | No Context      | 0.07           | 0.02             | 0.05         | 0.00         | 0.01       | 0.00        | 0.03  |
> | Text ICL xBase    | 0.25           | 0.06             | 0.16         | 0.01         | 0.15       | __0.03__    | 0.11  |
> | Text ICL xPatch    | __0.62__       | __0.23__         | __0.47__     | __0.11__     | __0.56__   | 0.02        |  **0.33** |
>
>
> [2] Odena et. al. Fuyu-8B: A Multimodal Architecture for AI Agents. 2023. https://www.adept.ai/blog/fuyu-8b.
>
> [3] Bai et. al. Qwen-VL: A Versatile Vision-Language Model for Understanding, Localization, Text Reading, and Beyond. arXiv 2023.

---

> > ### Author Response · Authors · 2024-11-22
> >
> > 4. *[T]he authors [...] claim that alignment with language is "not immediately obvious" for image ICL*
> >
> > **Prior work has shown that the input image and text embedding of VLMs are quite different, i.e., they have a very low cosine similarity [4] and separate into distinct clusters when visualized with PCA [5]**, therefore, it is not immediately clear that image ICL and text ICL are aligned. We have revised the main text to add references to these works. Due to the significant differences between the image and text inputs, we maintain that alignment is not obvious and that our finding that tokens evolve similarly for image and text ICL is an exciting contribution that motivates our proposed cross-modal patching.
> >
> > [4] Lin et. al. VILA: On Pre-training for Visual Language Models. CVPR 2024.
> >
> > [5] Liang et. al. Mixture-of-Transformers: A Sparse and Scalable Architecture for Multi-Modal Foundation Models. arXiv 2024.
> >
> > ---
> >
> > 5. *[Provide] quantitative results [for task conflict].*
> >
> > We include quantitative results for task conflict. We evaluate 100 random pairs of conflicting tasks derived from VQAv2. **The results indicate that patching instruction vectors (Instruction xPatch) is highly effective in steering the model toward a different task, outperforming the common alternative of including the instruction in the system prompt (System Prompt) by 27\%.** Additionally, we include a baseline (Instruction xBase) where the model is not provided with the new instruction. We include this table in Sec. A.5 of the Appendix.
> >
> > **Table 2. Instruction xPatch effectively steers the model to perform a newly introduced (and conflicting) task.**
> > | Method                                           | Acc.   |
> > |-------------------------------------------------|--------|
> > | Instruction xBase       | 0.04   |
> > | Instruction xBase + System Prompt                       | 0.05   |
> > | **Instruction xBase + Instruction xPatch** | **0.32** |
> >
> > ---
> >
> > 6. *[Provide] quantitative results [for image ICL transfer].*
> >
> > We quantify the performance of cross-modal patching applied to dense text descriptions, evaluating on 12 queries corresponding to Figure 10 of the main paper. **We show that image ICL examples are much more effective when patched (Image ICL xPatch) rather than few-shot prompted (Image ICL xBase), with a 17\% improvement. We also see that Image ICL xPatch outperforms text-based examples, either by prompting (Text ICL xBase) or by patching (Text ICL xPatch).** We include the following table in Sec. A.6 of the Appendix.
> >
> > **Table 3. Accuracy for transfer from image ICL to dense text descriptions.**
> > | Model             | Acc.   |
> > |-------------------|--------|
> > | No Context        | 0.00   |
> > | Text ICL xBase    | 0.17   |
> > | Text ICL xPatch    | 0.17   |
> > | Image ICL xBase    | 0.08   |
> > | **Image ICL xPatch**    | **0.25** |
> >
> > ---
> >
> > 7. *The paper does not include an Appendix.*
> >
> > We had uploaded the Appendix as a separate PDF under “Supplementary Material” in our original submission. Thank you for pointing out the typo in our reference of Table 4; we have revised this in the manuscript.

---

> > > ### Author Response · Authors · 2024-11-25
> > > **Response to Reviewer oiXL**
> > >
> > > We are bumping this thread, since we have not yet received a response from Reviewer oiXL. We would like to highlight for Reviewer oiXL that we have added the suggested quantitative results to the manuscript. **Specifically, we have added Sec. A.3, A.5, A.6 to the Appendix (which can be found as a separate pdf under “Supplementary Material”) and updated Sec. 2.2 of the main paper, which has improved the paper with the help of your feedback.** We invite Reviewer oiXL to review these updates and consider increasing your score if your concerns have been adequately addressed.

---

### Author Response · Authors · 2024-11-27
**Response to All Reviewers**

Dear Reviewers,

We greatly appreciate the time and effort you have invested into your initial reviews. It would mean a great deal to us if you could kindly review our responses **before the discussion period concludes next week on December 2nd.**

Best Regards,

The Authors

---

### Meta-Review · Area_Chair_dvBB · 2024-12-21

**Metareview:**

This paper investigates how vision-and-language models (VLMs) encode task representations across modalities, demonstrating that conceptually similar tasks map to similar vector representations regardless of input modality. The authors propose methods for transferring task vectors between modalities and combining instruction-based and exemplar-based approaches.

### Strengths:
1. Novel investigation of cross-modal task representations
> "The paper introduces cross-modal task vectors, demonstrating that VLMs can generalize tasks across different input modalities (e.g., text to image) effectively,advancing multi-modal interpretability"

2. Systematic empirical evaluation with quantitative results
> "The paper reveals a consistent three-stage token evolution (input, task, answer) across modalities, providing deeper understanding of VLM internal mechanisms"

3. Technical soundness with thorough experimental validation
> "The experiments seem to fully explore the task-space in the domain of these llava-like models"

### Weaknesses:
1. Lack of methodological clarity and reproducibility
> "The methodology is straightforward. Although the authors mention that task vectors are cross-modal and can transfer between different modalities, this work appears to be an extension of previous work on function vectors. The method seems like a direct extension, which is trivial and lacks novelty"

2. Limited technical novelty beyond existing work
> "Patching itself is never actually described, not clear how the proposed method actually works at test time. Are there any hyper-parameters used? I know when I run model steering vectors, I'll normalize the vector and add it by 3x, is that happening here?"

3. Insufficient experimental rigor and validation
> "For several experimental results, such as task conflict and image ICL transfer, the paper primarily presents qualitative examples as evidence. Providing quantitative results would strengthen the claims"


### Justification:
The paper, while exploring an interesting direction in cross-modal task representations, falls short of ICLR standards for several reasons:

1. The technical contribution appears to be an incremental extension of existing work on function vectors, without sufficient novel insights or theoretical advances.

2. Despite revision attempts, core methodological details remain unclear, hampering reproducibility.

3. The experimental validation, while improved with additional quantitative results, still relies heavily on qualitative examples for key claims.

While the authors made commendable efforts to address reviewer concerns, the fundamental issues around novelty and technical depth suggest this work would benefit from substantial development before being ready for publication at ICLR.

**Additional Comments On Reviewer Discussion:**

While the authors made efforts to address reviewer concerns by adding quantitative results and clarifying methodological details, fundamental issues remain:

1. Two reviewers (oiXL and x3D1) gave detailed critiques about lack of novelty and unclear methodology. Though authors added more experiments, the core concerns about technical innovation were not fully addressed.

2. The third reviewer (23Jk) was more positive but still noted significant methodological gaps.

3. None of the reviewers engaged with author responses during discussion, despite authors' multiple attempts to address concerns.

---

### Decision · Program_Chairs · 2025-01-22

Reject